# Evidence of Coulomb liquid phase in few-electron droplets

Jashwanth Shaju[1,4], Elina Pavlovska[2,4], Ralfs Suba[2], Junliang Wang[1], Seddik Ouacel[1], Thomas Vasselon[1], Matteo Aluffi[1], Lucas Mazzella[1], Clément Geffroy[1], Arne Ludwig[3], Andreas D. Wieck[3], Matias Urdampilleta[1], Christopher Bäuerle[1], Vyacheslavs Kashcheyevs[2✉] & Hermann Sellier[1✉]

Emergence of universal collective behaviour from interactions within a sufficiently large group of elementary constituents is a fundamental scientific concept[1]. In physics, correlations in fluctuating microscopic observables can provide key information about collective states of matter, such as deconfined quark–gluon plasma in heavy-ion collisions[2] or expanding quantum degenerate gases[3,4]. Mesoscopic colliders, through shot-noise measurements, have provided smoking-gun evidence on the nature of exotic electronic excitations such as fractional charges[5,6], levitons[7] and anyon statistics[8]. Yet, bridging the gap between two-particle collisions and the emergence of collectivity[9] as the number of interacting particles increases[10] remains a challenging task at the microscopic level. Here we demonstrate all-body correlations in the partitioning of electron droplets containing up to $N = 5$ electrons, driven by a moving potential well through a Y-junction in a semiconductor device. Analysing the partitioning data using high-order multivariate cumulants and finite-size scaling towards the thermodynamic limit reveals distinctive fingerprints of a strongly correlated Coulomb liquid. These fingerprints agree well with a universal limit at which the partitioning of a droplet is predicted by a single collective variable. Our electron-droplet scattering experiments illustrate how coordinated behaviour emerges through interactions of only a few elementary constituents. Studying similar signatures in other physical platforms such as cold-atom simulators[4,11] or collections of anyonic excitations[8,12] may help identify emergence of exotic phases and, more broadly, advance understanding of matter engineering.

Breaking up matter into pieces and studying the statistics of fragments is one of the basic epistemic strategies in physics. Arguably the most notable pursuit of this strategy is the success of high-energy particle colliders in discovering and quantifying the fundamental types of matter within the Standard Model of elementary particles. In the studies of strong interaction, relativistic ion collisions are used to induce the deconfinement of the nuclear matter (composed of correlated hadrons) into a hot plasma of more fundamental particles (quarks and gluons)[13]. The statistical fluctuations of collision products carry rich information about the collective dynamics[9]. In particular, measurements of high-order cumulants have been used[2,14] to pinpoint the critical point in the phase diagram of quantum chromodynamics (QCD)[15,16]. In solid-state nanoelectronic circuits, charged quasiparticles can be launched with on-demand single-electron sources and guided to a small interaction area such as a quantum point contact (QPC)[7,17] or an energy barrier[18–20], creating a collider analogue for electronic matter. Second-order correlations in steady-state and on-demand collisions have provided an essential tool to decode partitioning noise of composite particles[5,6], fermionic[7,17] and anyonic[8] exchange statistics and two-particle Coulomb interactions[18–20]. In previous research, temporal electronic correlations in nanostructures have been extensively studied[21–27]. These experiments primarily investigated the Coulomb interaction between two neighbouring electrons as they traverse a QPC, quantum dot (QD) or tunnel junction.

Higher-order ($k > 2$) correlations in current fluctuations[28,29] and electron-counting statistics[25,30,31] have been recognized as important signatures of Coulomb interactions. Yet, evidence for the corresponding collective behaviour in on-chip transport has been difficult to interpret[32] owing to limited control over the number $N$ of interacting particles and the dominating randomness of tunnelling times. Investigating the gap between few-particle correlations and the thermodynamic limit[10] for 2D electron systems is motivated by their rich phase diagram as a function of electron density, temperature and magnetic field, including strongly correlated Coulomb liquid, Wigner crystals and quantum Hall phases[33].

Here, by drawing an analogy with relativistic ion collisions, we investigate partitioning of a small electron-plasma droplet containing a precise number $N$ of electrons. Analysing the partitioning data of our

[1]Université Grenoble Alpes, CNRS, Grenoble INP, Institut Néel, Grenoble, France. [2]Department of Physics, University of Latvia, Riga, Latvia. [3]Lehrstuhl für Angewandte Festkörperphysik, Ruhr-Universität Bochum, Bochum, Germany. [4]These authors contributed equally: Jashwanth Shaju, Elina Pavlovska. ✉e-mail: slava@latnet.lv; hermann.sellier@neel.cnrs.fr

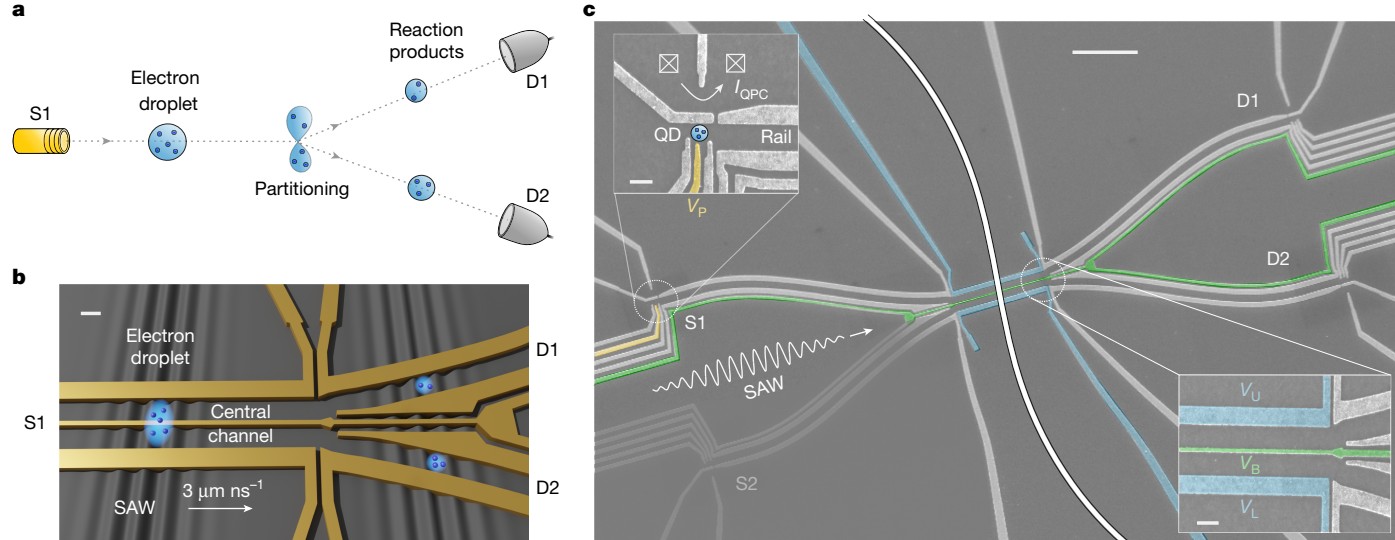

**Fig. 1 | Partitioning of an electron droplet. a**, Schematic of the experiment. An electron source (S1) delivers a few-electron droplet, which is split in-flight at a Y-junction. The output of the partitioning is analysed by two single-shot detectors (D1 and D2). **b**, Schematic of the electron droplet transport inside the selected potential minimum of a SAW. Electrostatic gates (yellow) are used to guide the electron droplet and create a Y-junction. Scale bar, 200 nm. **c**, Scanning electron microscope image of the device showing the metallic surface gates (light grey). The electron source (S1) consists of a QD (shown in the top-left inset) coupled to a QPC for charge sensing. The plunger gate (yellow) is used to inject a precise number of electrons into a single SAW minimum. A second electron source (S2) is connected to the central channel to inject more electrons. The Y-junction at the end of the central channel (see bottom inset) enables partitioning of the electron droplet. Scale bars, 2 μm (main); 200 nm (insets).

synthesized electron droplet using multivariate cumulants enables us to identify its corresponding strongly correlated state of electronic matter.

## On-chip multi-electron splitter

We have implemented the partitioning of a charge droplet of interacting electrons using a Y-junction in a GaAs semiconductor heterostructure, as illustrated in Fig. 1. Two single-electron sources and two single-shot detectors are made of gate-defined QDs paired with nearby QPCs used as charge sensors. By recording the QPC current $I_{QPC}$ of each QD before and after the experiment, the precise number of released electrons (source QD) and captured electrons (detector QD) is measured. Several pairs of parallel electrodes define depleted quasi-1D transport rails, guiding the electrons from the sources to the detectors. A 40-μm-long central channel is used to control the electron droplet properties before partitioning. This channel includes a narrow 30-nm-wide barrier gate that enables precise tuning of the confining potential in the direction perpendicular to the rail. At the end of the channel, a Y-junction splits the electron droplet into two parts and directs the 'reaction products' towards the detectors. The counting statistics is accumulated into the probabilities $P_{(N-n,n)}$, in which $n$ and $N-n$ are the numbers of electrons measured, after each single-shot partitioning, in detectors D1 and D2, respectively.

In our experiment, the electron droplet is transported within a single piezoelectric potential minimum of a surface acoustic wave (SAW)[34–36]. An interdigital transducer (IDT), positioned 1.5 mm away, generates a 180-μm-long SAW train. By applying a voltage pulse $V_P$ on the plunger gate of the source QD, with a duration much shorter than the SAW period, a well-defined number of electrons (ranging from 1 to 5) can be loaded into a single minimum of the SAW potential (see Supplementary Note 1). When the SAW propagates across the device, these electrons remain confined in the moving QD[37], which shuttles them along the rails.

For droplets with more than three electrons, the injection from a single source becomes technically challenging and we instead prepare these droplets using two sources, synchronized to the same SAW minimum. The two parts then merge in the central channel, in which the voltage $V_B$ on the barrier gate is tuned to ensure that the electrons lose their history and become statistically indistinguishable (see Supplementary Note 3).

## Partitioning of an electron droplet

We illustrate our ability to control the partitioning in Fig. 2, in which the counting statistics $P_{(N-n,n)}$ is shown for $N=4$ as a function of the voltage difference $\Delta = V_U - V_L$ between the two electrodes defining the central channel. This parameter acts as a tunable impact parameter for the collision with the Y-junction.

The simplest case is when all of the electrons are placed in different SAW minima (Fig. 2a) such that they are prevented from forming a droplet and cannot interact. We find that the counting statistics $P_{(N-n,n)}$ can be reconstructed from single-electron partitioning data (solid lines) and thus follows a binomial distribution, with electrons scattering at the Y-junction independently of each other. Such statistics corresponds to fixed-$N$ samples of a non-interacting electron gas.

To induce correlations, we group the electrons in two pairs, placed in adjacent SAW minima (Fig. 2b). An increase of the probability $P_{(2,2)}$ can be observed compared with the non-interacting case, indicating antibunching of the two electrons contained in each pair[18]. To obtain a strongly correlated state, we place all four electrons in the same SAW minimum (Fig. 2c) and note a similar increase in $P_{(2,2)}$ but the maxima of $P_{(1,3)}$ and $P_{(3,1)}$ now exceed $P_{(2,2)}$. Although the probabilities in Fig. 2b,c are qualitatively different, the multi-electron interdependencies are difficult to interpret directly from the counting statistics.

## Multivariate cumulants

To interrogate the nature of the many-electron state in the droplet, we aim to characterize its internal correlations and decompose them into irreducible components, known as cumulants[38]. Cumulants are convenient as they capture not only pairwise but also higher-order

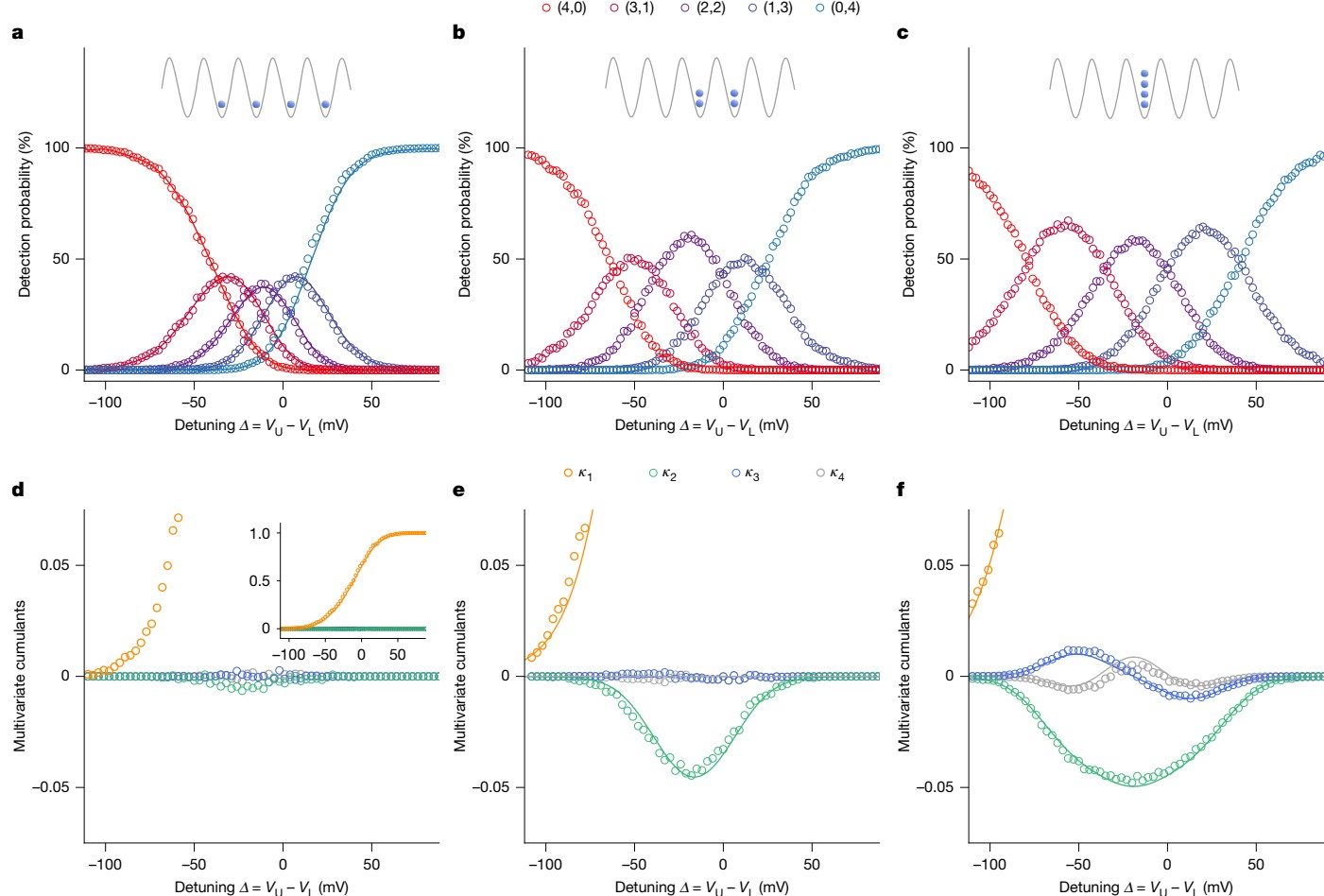

**Fig. 2 | Partitioning of an electron droplet containing N = 4 electrons.**
**a**–**c**, Detection probabilities $P_{(N-n,n)}$ versus side-gate detuning voltage $\Delta$. Each data point is extracted from 3,000 single-shot measurements. Error bars on probabilities are smaller than the symbol size (see Methods). The labels $(N - n, n)$ correspond to the events in which $n$ electrons are measured at detector D1 and $N - n$ electrons at detector D2. In **a**, the four electrons are distributed across different SAW minima, as illustrated in the top inset. In **b**, the four electrons are loaded into adjacent minima, with two electrons in each. In **c**, all four electrons are confined into a single minimum. The solid lines in **a** are predictions based on independently measured single-electron partitioning probabilities (see Supplementary Note 4). **d**–**f**, Multivariate cumulants $\kappa_1...\kappa_N$ calculated from the measured probabilities shown in **a**–**c**. The inset in **d** shows the evolution of $\kappa_1$ across the entire range and the solid line is the partitioning probability $P_{(0,1)}$ of a single electron. In **e**, two non-equivalent cumulants contribute to $\kappa_2$ (see Methods and Supplementary Note 6). Solid lines in **e** and **f** are fits using the Ising model of equation (1).

correlations. This is crucial for understanding complex many-body systems in which strong enough pairwise interactions can lead to correlations of all orders, heralding the emergence of a new collective state. One possibility is to consider the high-order cumulants $\langle\!\langle n^k \rangle\!\rangle$, or their combinations such as skewness and kurtosis, of the collective variable $n$, as it is measured directly[2,31]. Yet, in this representation, contributions of individual particles are not resolved. To explain few-electron correlation effects, it is crucial to separate these correlations by order, corresponding to the number of particles involved. We achieve this by recognizing that $n = T_1 + T_2 + ... + T_N$ is a sum of several variables $T_j$, corresponding to the partitioning outcome of each electron ($T_j = 1$ or 0 if the $j$th electron is detected at D1 or D2, respectively). Instead of $\langle\!\langle n^k \rangle\!\rangle$, we consider the irreducible correlation functions $\langle\!\langle T_i T_j ... T_k \rangle\!\rangle$, known as multivariate cumulants in statistics[39] or connected diagrams in field theory[40], to quantify the effect of interactions. Notably, if the presence of the $i$th electron does not influence the $j$th electron, all multivariate cumulants involving both $T_i$ and $T_j$ will be zero. Here we focus on the symmetrized multivariate cumulants $\kappa_k$ defined by averaging the cumulants over all possible combinations of exactly $k$ distinct variables $T_j$ out of $N$. If electrons are statistically indistinguishable (all placed in the same SAW minimum or all in different SAW minima), all terms in the averaging are equal and $\kappa_k = \langle\!\langle T_1 T_2 ... T_k \rangle\!\rangle$. In this case, the multivariate cumulants $\kappa_k$ are

entirely determined by the counting probability distribution $P_{(N-n,n)}$ and can be computed from both measurements and models (see Methods).

We now illustrate the meaning of multivariate cumulants using the experimental data from Fig. 2a–c (in which $N = 4$) and show the corresponding cumulants ($\kappa_1...\kappa_4$) in Fig. 2d–f. The first-order average $\kappa_1 = \langle T_1 \rangle = \langle n \rangle / N$ is simply the marginal probability for one electron to be transmitted into D1; it changes monotonously from 0 to 1 with detuning parameter $\Delta$. For a binomial distribution of independent trials, all high-order cumulants $\kappa_{k>1}$ are zero, and this is indeed the case in Fig. 2d in which all four electrons are distributed into separate SAW minima. To gain intuition about the second-order correlations, we consider the second central moment $\langle n^2 \rangle - \langle n \rangle^2$ of the counting statistics, which is always equal to $N\kappa_1(1 - \kappa_1) + N(N-1)\kappa_2$. It consists of two terms: the ideal gas contribution proportional to $N$ and representing independent shot-noise accumulation, and the interaction-driven term proportional to $\kappa_2$. Coulomb repulsion leads to negative two-body correlations and the corresponding suppression of fluctuations in $n$ (antibunching). $\kappa_2 = \langle\!\langle T_1 T_2 \rangle\!\rangle < 0$ means the choice, that one electron makes, tends to be opposite to what the other electrons do. Indeed, we observe $\kappa_2 < 0$ in both Fig. 2e for two electron pairs and Fig. 2f for a quadruplet. The difference between the two cases is revealed by considering higher orders of correlation: although $\kappa_3$ and $\kappa_4$ are close to zero when electrons are

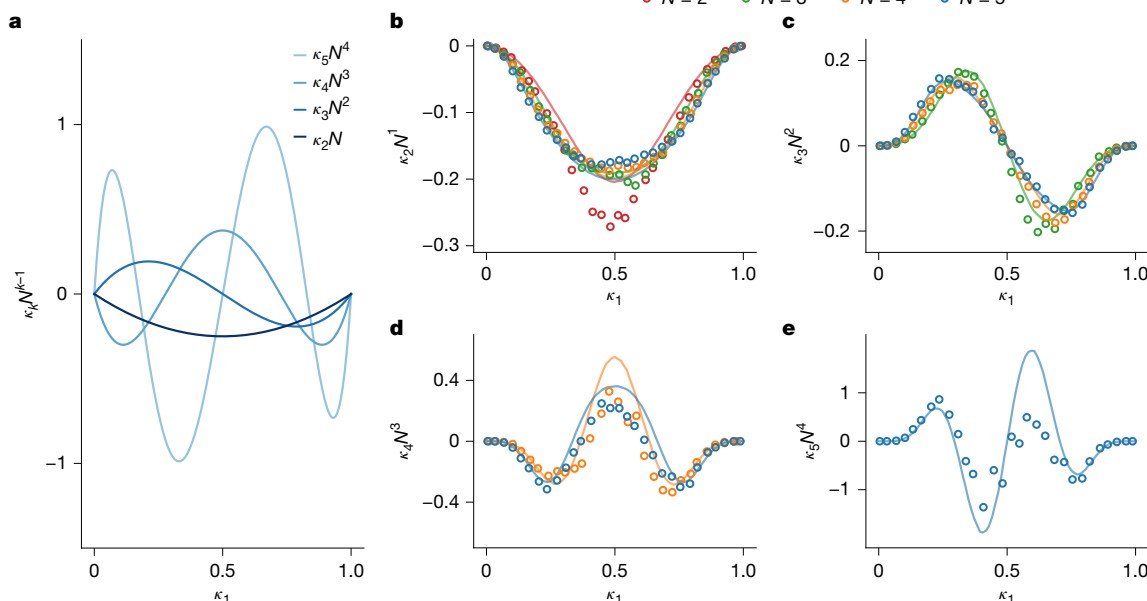

**Fig. 3 | Scaling of correlation functions with the number of particles.**
**a**, Leading-term universal asymptotics of the repulsion-dominated cumulants in the large-$N$ limit, as a function of $\kappa_1$ according to equation (11). **b–e**, Measured cumulants $\kappa_k$ of order $k = 2$–$5$ for droplets with $N = k \ldots 5$. Lines show corresponding simulations of sudden partitioning of an equilibrium Coulomb plasma confined in a quartic parabolic potential with realistic microscopic parameters (see Methods).

restricted to interact in pairs only (Fig. 2e), all cumulants $\kappa_k$ up to $k = N$ are generally non-zero when all $N$ electrons are placed in the same SAW minimum (Fig. 2f for $N = 4$ and Extended Data Fig. 2 for $N = 3$–$5$). Higher-order cumulants oscillate with detuning and exhibit $k - 1$ extrema separated by $k - 2$ zeros. Even cumulants are symmetrical whereas odd cumulants are antisymmetrical, reflecting the symmetry of the Y-junction already evident from the partitioning probabilities in Fig. 2a–c.

In the following, we show that the observed pattern of high-order correlations aligns with the universal signature of a strongly correlated liquid and locate the droplet state within the phase diagram associated with the gas–liquid crossover.

## Universal signatures of a Coulomb liquid

The relevant state of matter for our electron droplets is a one-component Coulomb plasma[41], which can undergo a temperature-driven crossover from a Coulomb gas at $T > T_c$ to a Coulomb liquid at $T < T_c$. The crossover temperature $T_c$ is determined by the competition between entropy and Coulomb repulsion energy and is characterized by the dimensionless plasma parameter $\Gamma^{(\text{pl})} \approx T_c/T$. In more conventional transport experiments, in which statically confined electrons are connected to external reservoirs, transition to a Coulomb liquid can manifest itself as an energy gap on the order $k_B T_c/N$ for particle addition (Coulomb blockade in QDs) or as temperature saturation of compressibility of a two-dimensional electron gas (2DEG)[42].

Here we rely solely on the finite-$N$ counting statistics to estimate the state of our Coulomb plasma, building an analogy with relativistic ion collisions used to study the phase diagram of QCD. In particular, at low baryonic densities[43], the transition from quark–gluon plasma at temperatures $T > T_c^{\text{QCD}}$ to hadronic fluid at $T < T_c^{\text{QCD}}$ is not a sharp phase transition but rather a smooth crossover[44,45]. Freeze-out of fluctuations (owing to quench of equilibrium during expansion[16]) determines the cumulants in the number of produced hadrons, which have been used to estimate $k_B T_c^{\text{QCD}} \approx 170$ MeV (ref. 2). Unlike in QCD, in which particles are created from the vacuum, the equilibrium ensemble for our Coulomb droplets is canonical, as the number of electrons $N$ is conserved in collisions with the Y-junction.

In the ideal gas limit of a Coulomb plasma, $\Gamma^{(\text{pl})} \ll 1$, the multivariate cumulants $\kappa_k$ take universal (albeit trivial) values, $\kappa_{k>1} \to 0$. In the opposite limit of an interaction-dominated liquid, $\Gamma^{(\text{pl})} \gg 1$, we can derive a universal form of the large-$N$ scaling of the cumulants $\kappa_k$ from the condition that the distribution of $n$ is governed by Coulomb repulsion rather than by the statistical fluctuations. For interactions to dominate the variance of the observable $n$, the interaction term must asymptotically cancel the ideal gas term, hence $\kappa_2 \to -\kappa_1(1 - \kappa_1)N^{-1}$. Extending the argument by induction to higher $k$, we find that $\kappa_k \propto N^{-k+1}$, with the proportionality coefficient given by a specific parameter-free polynomial of order $k$ in $\kappa_1$, as shown in Fig. 3a (see Methods). In statistical physics, the liquid phase is characterized by low compressibility, as it resists changes in particle number. Thus the strongly correlated limit, derived here from the condition of vanishing fluctuations in the thermodynamic limit, $\langle\!\langle n^2 \rangle\!\rangle/N \to 0$, corresponds to an incompressible fluid.

As our experiment allows the tuning of $\kappa_1$, we plot in Fig. 3b–e the rescaled cumulants $\kappa_k N^{-k+1}$ as functions of $\kappa_1$ for $k$ up to 5, using our experimental data for $N = k \ldots 5$. We observe that the scaling with $N$ expected in the interaction-dominated limit is obeyed as soon as $N \geq 3$. The pattern and magnitude of oscillations in Fig. 3b–e are also in good qualitative agreement with the universal prediction in Fig. 3a, confirming that our droplets are large enough to exhibit the emergent behaviour of a strongly correlated liquid.

## Effective Ising model

For quantitative analysis of our finite-$N$ data in terms of interaction strength and thermodynamic phase diagram, we use the archetype of classical lattice gas models[46] and describe the gas–liquid transition with the Ising model on a complete graph (all-to-all interactions). This model is defined by the following Hamiltonian, expressed in terms of directly measurable partitioning variables,

$$\mathcal{H} = U \sum_{\substack{i,j=1 \\ i \neq j}}^{N} \left(T_i - \frac{1}{2}\right)\left(T_j - \frac{1}{2}\right) + \mu \sum_{i=1}^{N} T_i \tag{1}$$

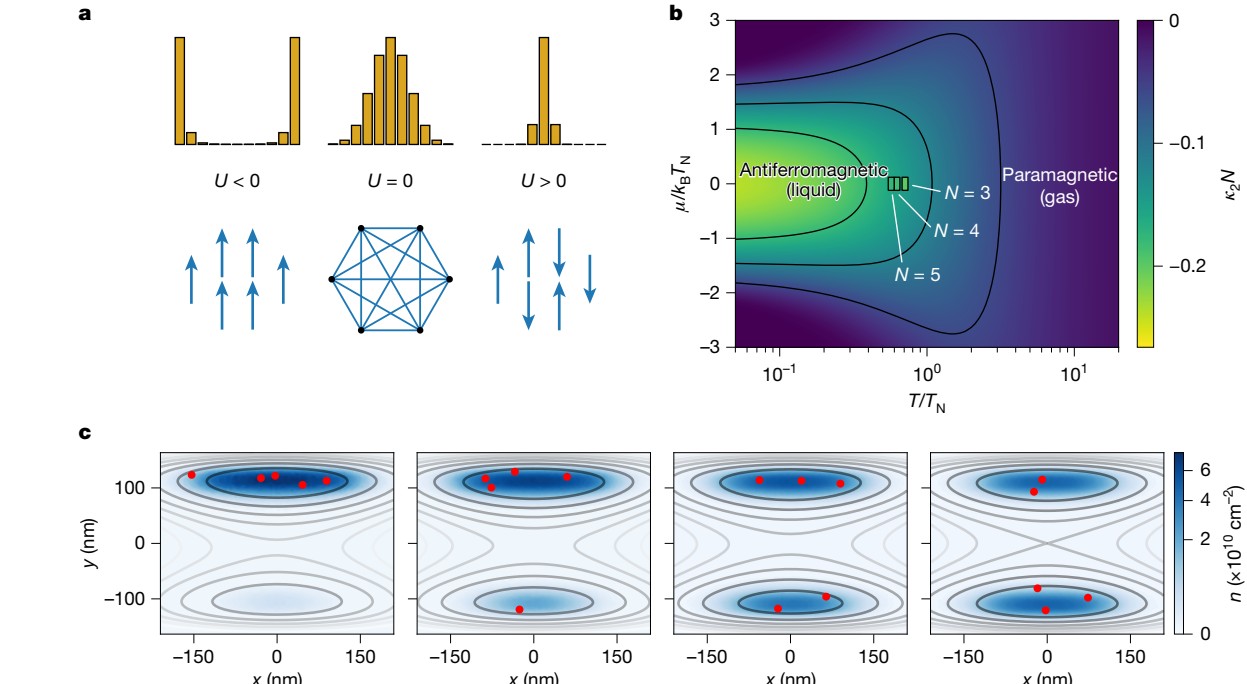

**Fig. 4 | Effective Ising model and Monte Carlo simulation of a Coulomb liquid droplet. a**, Interpretation of partitioning in terms of magnetic spin interactions. Uncorrelated partitioning ($U = 0$, binomial distribution in the middle), bunching ($U < 0$) and antibunching ($U > 0$) correspond, respectively, to paramagnetic, ferromagnetic and antiferromagnetic phases of the Ising model on a complete graph, for which counting statistics gives the distribution of the total magnetization. **b**, Phase diagram of the antiferromagnetic crossover in the thermodynamic limit of the Ising model, with appropriately scaled negative pair correlations $\kappa_2 N$ as the order parameter. The axes are given by the temperature $T$ and the magnetic field $\mu$, scaled by the Néel temperature $T_N$. The measured correlations for $N = 3$, 4 and 5 at $\mu = 0$ are shown by colours in small

squares. The horizontal position $T/T_N$ of the squares is obtained from the fits of the Ising model to the partitioning curves (Extended Data Table 1). The slight deviations in colour between the phase diagram and the measured values (36%, 20% and 11%, respectively) are dominated by the finite-$N$ effect, not by discrepancy with the model. **c**, Four configurations of the 2D confining potential in the central channel (level lines), together with snapshots of the spatial positions of $N = 5$ electrons (red dots) from Monte Carlo simulations of a classical Coulomb plasma (see Methods). The colour scale shows the calculated average electron density. The four panels correspond to (from left to right) $\Delta - \Delta_0 = 101$, 60, 21 and 0 mV.

in which $U$ is the interaction strength and $\mu$ controls the upper–lower charge balance within the central channel, with $\mu = 0$ corresponding to symmetric partitioning statistics $P_{(N-n,n)} = P_{(n,N-n)}$. We find that a sudden quench of equilibrium fluctuations governed by the Ising Hamiltonian (equation (1)) at temperature $T$ accurately reproduces the measured cumulants as a function of $\mu = -\alpha(\Delta - \Delta_0)$, as shown by solid lines in Fig. 2e,f for $N = 4$ and in Extended Data Figs. 1 and 2 for $N = 2$–5. In this model, $U/k_B T$ and $\Delta_0$ are fitted independently for each $N$, whereas $\alpha/k_B T$ is fixed globally ($\alpha$ is the voltage-to-energy conversion factor). Fitted values are listed in Extended Data Table 1.

The Ising model establishes a useful analogy between the phases of magnetically interacting spins $s_i = 2T_i - 1 = \pm 1$ and our partitioning statistics (Fig. 4a). $U$ is the energy cost for two spins to be parallel (for two electrons to exit on the same side of the Y-junction), with a positive $U$ making configurations with opposite spins preferable (antiferromagnetic coupling). As the Coulomb plasma is characterized by repulsive interactions ($U > 0$), the gas-to-liquid transition corresponds to the paramagnetic-to-antiferromagnetic transition of Ising spins. Unlike the usual sharp phase transition in the ferromagnetic case ($U < 0$), here it is a crossover in temperature because an all-to-all antiferromagnetic coupling ($U > 0$) imposes global correlations but not a particular spin pattern. In the large-$N$ limit and in the absence of polarizing field ($\mu = 0$), the characteristic scale of the antiferromagnetic crossover is the Néel temperature $k_B T_N = UN/2$ (see Methods), which we identify with the crossover temperature $T_c$ of the strongly correlated Coulomb liquid.

The cumulants $\kappa_k$ serve as the irreducible correlation functions of the Ising spins and can be used as order parameters to quantitatively trace the crossover in a phase diagram. This is illustrated for $\kappa_2 N$ in Fig. 4b,

in which the scale $T_N$ represents the characteristic energy required to destroy the correlated state, by either thermal fluctuations ($T$ axis) or by an external field favouring ferromagnetic order ($\mu$ axis). Converting the fitted parameters $U/(k_B T)$ to $T/T_N$ for each $N$ (see Extended Data Table 1) allows us to compare the measured correlations with their values in the thermodynamic limit. In terms of plasma state, we find the results to be closer to liquid than gas ($T < T_N$) for $N = 3$–5, consistent with the observation of cumulant scaling (see Fig. 3) characteristic of the liquid state.

Finally, we complement the coarse-grained effective Ising model with an explicit microscopic simulation of the Coulomb plasma[47] under the specific conditions of our experiment. Figure 4c illustrates the case of $N = 5$ electrons, interacting through an unscreened Coulomb potential and placed in a quartic-parabolic confining potential for different values of detuning voltage $\Delta$. The colour represents the canonical probability density drawn from classical Monte Carlo simulations and the red dots represent a particular snapshot of the corresponding electron positions. The simulated partitioning cumulants are shown in Fig. 3b–e with continuous lines. As the shape of the confinement potential is computed from an electrostatic modelling (see Methods), the only adjustable parameter is the effective temperature of the droplet, $T = 25$ K (consistent with other estimates; see Supplementary Note 2). We observe that these microscopic simulations are consistent with the observed deviations from the ideal scaling and account for finite-size and temperature effects. We have verified that the qualitative agreement with the scaling limit (Fig. 3a) is robust to the choice of the confinement potential and to the competition of Coulomb and exchange correlations, as long as the plasma parameter is large enough.

## Conclusion and outlook

Inspired by relativistic ion colliders used to study quark–gluon plasma, we have successfully created a plasma droplet of strongly correlated electrons on a microchip. Confining electrons with a SAW enables precise manipulation of the number of interacting particles within the electron-plasma droplet and a gate-tunable Y-junction provides deterministic control of the effective impact parameter.

A multivariate cumulant analysis of the partitioning data reveals the formation of a strongly correlated Coulomb liquid, emerging with as few as three electrons in the droplet. This mirrors the incompressible liquid of hadrons bound by nuclear forces, but—unlike high-energy ion collisions—our approach allows us to trace universal signatures of collectivity at very low particle numbers.

Drawing a powerful analogy with condensed-matter physics, our results align well with an Ising model on a complete graph. The observed electron antibunching, governed by Coulomb interactions, can be compellingly interpreted as antiferromagnetic ordering below the Néel temperature.

In future, an exciting direction would be to extend this methodology to lower effective temperatures and strong magnetic fields, for which quantum Hall states emerge in 2D electron systems[33] and have already been simulated for small particle numbers[48]. Notably, evidence of electron bunching in a pair-partitioning experiment[49] under high magnetic fields suggests the potential formation of a Laughlin state droplet[50], opening new avenues for engineering exotic correlated states in electron systems.

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

## Methods

### Device description

The device is fabricated in a Si-doped GaAs/AlGaAs heterostructure grown by molecular-beam epitaxy. The 2DEG resides 110 nm below the surface, with electron density $2.8 \times 10^{11}\,\mathrm{cm^{-2}}$ and mobility $9 \times 10^5\,\mathrm{cm^2\,V^{-1}\,s^{-1}}$. Metallic gates (Ti, 3 nm; Au, 14 nm) are deposited on the surface of the semiconductor using electron-beam lithography. All measurements are performed at a temperature of about 20 mK in a $^3$He/$^4$He dilution refrigerator. The sample and measurement scheme are the same as in ref. 18. A set of negative gate voltages is applied to the surface gates to deplete the 2DEG underneath and create the nanostructures, including four QDs, four QPCs and two guiding rails, which are fully depleted. These rails connect the source QDs to the detector QDs and merge in the centre to form a single 40-μm-long channel, equipped with a narrow barrier gate in the middle to tune the shape of the confining potential from a single well to a double well.

### SAW generation

The SAW is generated using a double-finger IDT deposited on the surface and placed at a distance of 1.5 mm from the device. The metallic fingers are fabricated using electron-beam lithography and thin-film evaporation (Ti, 3 nm; Al, 27 nm) on the heterostructure. The IDT consists of 111 cells with a periodicity of 1 μm and a resonance frequency of 2.86 GHz at low temperature. The aperture of the transducer is 50 μm. To perform electron transport by SAW, a radiofrequency signal is applied on the IDT at its resonance frequency for a duration of 60 ns. To have a strong SAW confinement potential, the signal is amplified to 28 dB using a high-power amplifier before being injected into a coaxial line of the cryostat through a series of attenuators. The velocity of the SAW is 2,860 m s$^{-1}$.

### Electron transfer

Each single-shot experiment corresponds to the transfer of one or a few electrons from the source QDs to the detector QDs using the SAW as the transport carrier. To prepare a given number $N$ of electrons in a source QD, we use a sequence of fast voltage pulses to the channel gate and reservoir gate controlling the tunnel barriers of the QD. This sequence consists of three steps: initializing the QD, loading the electrons into the QD and preparing the QD for electron transfer. To initialize the source QD, electrons previously present in the QD are removed. Then, a given number of electrons are loaded into the QD by accessing a particular loading position in the charge-stability diagram of the QD. Finally, these electrons are trapped within the QD by switching to a holding configuration, from which they will be taken away by the SAW. At the same time, the two detector QDs are set in a configuration for which the electrons transported by the SAW will be captured with high fidelity. For more details, see Supplementary Note 1. By sensing the QPC currents of both the source and detector QDs with single-electron resolution and comparing their values before and after the electrons are transferred by the SAW, the precise number of electrons transferred to each detector can be determined. When calculating the partitioning probability, the very few events that do not conserve the total number of electrons are excluded by a post-selection routine.

In our experiment, the electrons are deterministically loaded into specific locations within the SAW train. The plunger gate of the QD is used to trigger the sending of the electrons into a precise minimum of the periodic SAW potential with a 30 ps resolution. This precise control allows for the formation of an electron droplet containing up to five electrons, using the two source QDs of the device. To synchronize the two trigger pulses with the radiofrequency signal generating the SAW, we use two arbitrary waveform generators combined with a synchronization module. The outputs of the arbitrary waveform generators are connected to the plunger gates by means of high-bandwidth bias tees for voltage pulsing and dc biasing.

### Electron partitioning

The electron droplet is partitioned at the Y-junction located at the end of the central channel, after a flight time of 14 ns. By applying a voltage detuning $\Delta = V_U - V_L$, in which $V_U$ and $V_L$ are the voltages applied to the side gates of the central channel, we can control the partitioning ratio between the two detectors D1 and D2. For all partitioning experiments reported here, the barrier gate voltage is set to $V_B = -1.25$ V to have a single central channel with a weak double-well potential profile. Careful analysis of the double-well potential and the electron number equilibration in the central channel is described in Supplementary Notes 2 and 3.

### Statistical uncertainty

The error in estimating the probability $p_n$ from the counting statistics is dictated by the distribution of independent Bernoulli trials. The corresponding likelihood function of measuring exactly $N_n$ outcomes of $n$ particles in detector D1 out of $N_{rep}$ repetitions is $\binom{N_{rep}}{N_n} p_n^{N_n}(1-p_n)^{N_{rep}-N_n}$. We use the mean $N_n/N_{rep}$ as the statistical estimate for $p_n$. The boundaries of the corresponding confidence interval (CI) at confidence level $1 - c = 95\%$ are determined from the likelihood function by inverting a one-tailed binomial test at significance level $c/2$, for the lower bound and the upper bound separately. At the extreme count $N_n = 0$ ($N_n = N_{rep}$), the lower (upper) boundary is set to 0 (1) and the other boundary is computed at significance level $c$. These CIs of probability measurements are used in the Ising model parameter estimation. For $0.06 < p_n < 0.94$ in our measurements with $N_{rep} = 3,000$, the CIs are almost symmetric and approximately given by $\pm 1.96\sqrt{p_n(1-p_n)/N_{rep}}$. Because the maximum width of the CI is about 0.036 (at $p_n = 0.5$), a value smaller than the size of the data points, we did not represent the error bars on the graphs of probabilities.

### Symmetrized multivariate cumulants

In our experiment, the observable is the number $n$ of electrons measured at detector D1, which can be expressed as a sum of binary variables $T_j$. From $n = \sum_{j=1}^{N} T_j$ and $T_j^2 = T_j$, we derive the general relation

$$m_k = \binom{N}{k}^{-1} \sum_{n=k}^{N} \binom{n}{k} p_n \tag{2}$$

between the probabilities $p_n = P_{(N-n,n)}$ of the full counting statistics (FCS)[21] and the $k$th-order symmetrized multivariate moments $m_k$ defined as averages of all permutations of $k$ distinct variables,

$$m_k = \binom{N}{k}^{-1} \sum_{1 \leq j_1 < j_2 < \ldots < j_k \leq N} \langle T_{j_1} T_{j_2} \ldots T_{j_k} \rangle, \tag{3}$$

in which $\binom{N}{k} = N!/[k!(N-k)!]$ is the binomial coefficient. The corresponding symmetrized multivariate cumulants

$$\kappa_k = \binom{N}{k}^{-1} \sum_{1 \leq j_1 < j_2 < \ldots < j_k \leq N} \langle\langle T_{j_1} T_{j_2} \ldots T_{j_k} \rangle\rangle \tag{4}$$

are, in general, not uniquely determined by FCS probabilities and their calculation requires further information (such as symmetry constraints or a microscopic model).

For statistically equivalent particles (that is, full permutational symmetry of the multivariate probability distribution), all terms in equations (3) and (4) are equal, and the moments $m_k$ can be related to cumulants $\kappa_k$ through standard univariate relations[51], $\ln(1 + \sum_{k=1}^{\infty} m_k z^k/k!) = \sum_{k=1}^{\infty} \kappa_k z^k/k!$. Using an explicit formula in terms of Bell polynomials[52], we can write

$$\kappa_k = \sum_{j=1}^{k} (j-1)!\,(-1)^{j-1} B_{kj}(m_1, m_2, \ldots, m_{k-j+1}). \tag{5}$$

See Supplementary Note 5 for the derivation of equation (2) and explicit formulas for $\kappa_k$ for $k = 1$–5. An example of correlated partitioning, in which equation (5) is not valid and the general combinatorial expressions for multivariate cumulants[39,53] need to be used, is shown in Fig. 2e and described in detail in Supplementary Note 6.

There is an important distinction between our method for extracting interaction signatures and the approach of so-called factorial cumulants considered in the context of electron transport[32] and particle physics[54]. The multivariate moments defined by equation (3) can be written $m_k = \langle (n)_k \rangle / (N)_k$, in which $(x)_k = x(x-1) \times \ldots \times (x-k+1)$ is the falling factorial and $\langle (n)_k \rangle$ is known as the factorial moment of FCS[32]. The $k$-dependent denominator $(N)_k$ in this expression for $m_k$ makes the $\kappa_k$ distinct from the factorial cumulants; see further discussion in Supplementary Note 5.

## Ising model on a complete graph

The Ising model on a complete graph is exactly solvable[55] and, hence, equilibrium fluctuations at any freeze-out quench temperature $T$ can be computed for any $N$. The Ising Hamiltonian of equation (1) can be expressed as a quadratic form of the observable $n = \sum_{j=1}^{N} T_j$,

$$\mathcal{H} = Un^2 + (\mu - NU)n + UN(N-1)/4. \tag{6}$$

The corresponding exact counting statistics in a canonical ensemble is $p_n = c_n/Z$ with the partition function $Z = \sum_{n=0}^{N} c_n$ and the statistical weights

$$c_n = \binom{N}{n} e^{-\beta Un(n-N) - \beta \mu n}, \tag{7}$$

in which $\beta = 1/k_B T$. Together with equations (2) and (5), this gives a way to calculate the exact multivariate cumulants $\kappa_k$ of all orders $k \leq N$ at any $N$.

To make the connection with the thermodynamic phase diagram in terms of $\mu$ and $T$ in the large-$N$ limit, explicit analytic expressions are obtained following ref. 55. We apply the lowest-order Stirling's formula $m! \approx m^m e^{-m} \sqrt{2\pi m}$ to the factorials in the binomial coefficient $\binom{N}{n}$ in equation (7) and perform expansion of $\ln(c_n)$ near its maximum $n \approx \langle n \rangle$ up to quadratic order. This results in a Gaussian approximation to $p_n$ of the form

$$p_n \propto e^{-(\beta + \beta')U(n - \kappa_1 N)^2}, \tag{8}$$

in which $\beta' = [4\kappa_1(1 - \kappa_1)k_B T_N]^{-1}$ and $k_B T_N = UN/2$ is the zero-field Néel temperature for the antiferromagnetic crossover. The relation between the effective magnetic field $\mu$ and the effective magnetization $\kappa_1 = \langle n \rangle / N$ in the large-$N$ limit is given by the transcendental equation[56]

$$2\kappa_1 - 1 = \tanh\left[ -\frac{T_N}{T}\left( 2\kappa_1 - 1 - \frac{1}{2}\frac{\mu}{k_B T_N} \right) \right], \tag{9}$$

which has only one solution for the antiferromagnetic sign of the coupling ($U > 0$).

To quantify the antiferromagnetic correlations in the thermodynamic limit, we choose the pair correlation function $\langle\langle T_1 T_2 \rangle\rangle = \kappa_2$ as the order parameter. It is obtained from the identity $\langle n^2 \rangle - \langle n \rangle^2 = N\kappa_1(1 - \kappa_1) + N(N-1)\kappa_2$ in the large-$N$ limit. Treating $n$ as a continuous variable and using the Gaussian approximation of equation (8), this gives the leading-order behaviour of $\kappa_2$ at a fixed $T/T_N$ and $N \to \infty$,

$$\kappa_2 N = -\frac{4\kappa_1^2(1 - \kappa_1)^2}{4\kappa_1(1 - \kappa_1) + T/T_N}. \tag{10}$$

A numerical solution to equation (9) together with equation (10) is used for the phase diagram in Fig. 4b.

As there is no lattice on a full graph favouring a particular pattern of staggered magnetization, the antiferromagnetic transition here is not a second-order phase transition but a crossover. The corresponding change in free energy has a weaker divergence ($\log N$) in the thermodynamic limit than at the ferromagnetic transition. The corresponding singular part[46] of the free energy change between $T = \infty$ and $T \to 0^+$ is $\beta\Delta F_{U>0} = (1/2)\ln(1 + T_N/T)$.

On the ferromagnetic side ($U < 0$), we note that $\kappa_2 N$ diverges when the temperature $T$ approaches the Curie temperature $T_C = -T_N > 0$ as $\kappa_2 N \propto (T - T_C)^{-1}$ for $T \to T_C^+$. At $T \leq T_C$, the Gaussian approximation of equation (8) breaks down and strong ferromagnetic order sets in. This corresponds to the droplet scattering at the Y-junction as a whole (without partitioning), with probability $\kappa_1$ to go to detector D1 and with $\kappa_2 = \kappa_1(1 - \kappa_1) > 0$ in the large-$N$ limit. For a large but finite droplet, there is no symmetry breaking, hence $\kappa_{k>1} = O(1)$, unlike $O(N^{-k+1})$ in the antiferromagnetic case.

## Universal scaling of partitioning cumulants

The interaction-dominated partitioning of a large droplet at $U > 0$ is described by the antiferromagnetic phase of the effective Ising model with $T/T_N \to 0$ and $N \to \infty$. The Boltzmann factor in equation (7) suppresses the fluctuations of $n$ around $\langle n \rangle = \kappa_1 N$ and caps the large-$N$ asymptotics of univariate cumulants from $\langle\langle n^k \rangle\rangle = O(N)$ (Gaussian limit of binomial distribution) to $\langle\langle n^k \rangle\rangle = O(1)$. From the latter condition (which is independent of the specifics of the Ising model), we derive the asymptotics $\kappa_k = G_k(\kappa_1)N^{-k+1} + O(N^{-k})$ for $k \geq 2$, in which the prefactor

$$G_k(\kappa_1) = -\frac{(k-1)!}{2} C_k^{(-1/2)}(2\kappa_1 - 1) \tag{11}$$

is universal and given by the ultraspherical (Gegenbauer) polynomials $C_k^{(a)}$ of degree $k$ and parameter $a = -1/2$. The first polynomials up to $k = 5$ are plotted in Fig. 3a, to show the universal strong-correlation asymptotics of the scaled cumulants $\kappa_k N^{k-1}$. Note that $G_2 = -\kappa_1(1 - \kappa_1)$ is also the zero-temperature limit of equation (10).

The polynomials $G_k(\kappa_1)$ have exactly $k - 2$ zeros for $0 < \kappa_1 < 1$, which explains the observed oscillation pattern and provides an exact specific example of oscillations in high-order cumulants[31]. We note that a similar generic $N^{-k+1}$ scaling has been discussed for cumulants of initial density perturbations in heavy-ion collisions[57], in which it arises for different reasons (dominance of autocorrelations in the independent point-sources model).

In contrast to antiferromagnetic correlations decaying with $N$ as $\kappa_k \propto N^{-k+1}$, the fluctuations in the ferromagnetic case are between $n = 0$ and $n = N$ only, hence $\kappa_k = O(1)$, and the limiting form for the unpartitioned scattering ($T/T_C \to 0$ in the Ising model) is the polynomial $\kappa_k = -\mathrm{Li}_{1-k}\left(\frac{\kappa_1}{\kappa_1 - 1}\right)$, in which Li is the polylogarithm.

## Coulomb liquid simulations

We model a finite droplet of Coulomb plasma in 2D using a confining single-electron potential $V_{1e}$ and an unscreened Coulomb potential[47], which results in the total potential

$$U(\mathbf{r}_1, \ldots, \mathbf{r}_N) = \sum_{i=1}^{N} V_{1e}(\mathbf{r}_i) + \sum_{i<j} \frac{e^2}{4\pi\epsilon_0\epsilon_r|\mathbf{r}_i - \mathbf{r}_j|}, \tag{12}$$

in which $\mathbf{r}_i = (x_i, y_i)$ is the in-plane coordinate of the $i$th electron and $\epsilon_r = 12.1$ is the relative dielectric permittivity in GaAs. The equilibrium distribution of electron coordinates is determined by a classical canonical ensemble at an effective temperature $T$. We sample electron positions $\{\mathbf{r}_i\}$ using a random walk Metropolis Monte Carlo algorithm designed to sample the canonical distribution. The convergence of the corresponding Markov chain is controlled by the Kolmogorov–Smirnov test[58]. For each set of parameters, a statistics of positions is collected with the estimated effective sample size ranging from $10^3$ to

$10^5$ depending on parameters. The statistics of positions is translated to partitioning statistics of a sudden quench using binary variables $T_i = \Theta(y_i)$, in which $\Theta$ is the Heaviside step function. This corresponds to an observable $n = T_1 + \ldots + T_N$ counting the number of particles in the $y > 0$ half-plane.

The confining electrostatic potential of our experiment can be approximated as a double-well quartic-parabolic 2D potential

$$V_{1e}(\mathbf{r}) = V_b + \mu_q \frac{y}{y_0} - 8V_b \frac{y^2}{y_0^2} + 16V_b \frac{y^4}{y_0^4} + \frac{m\omega_x^2 x^2}{2}, \tag{13}$$

in which $V_b$ is the height of the central barrier, $y_0$ is the distance between the two minima, $\mu_q$ is the transverse energy detuning proportional to the side-gates voltage difference $\Delta - \Delta_0$ (which controls the partitioning of the droplet) and $\omega_x$ is the oscillation frequency in the longitudinal direction, resulting from the confinement potential of the SAW.

The barrier height $V_b = 27.5$ meV and the distance between minima $y_0 = 220$ nm are estimated from an electrostatic simulation of the gate-controlled potential as explained in Supplementary Note 2. The transverse oscillation frequency in the two potential minima is then calculated as $\omega_y = \sqrt{32V_b/(my_0^2)} = 7.0$ THz using the effective mass $m = 0.067m_e$ for electrons in GaAs. The longitudinal oscillation frequency in the SAW potential is estimated from the peak-to-peak amplitude $A_{SAW} = 42$ meV and wavelength $\lambda_{SAW} = 1$ μm, using the relation $\omega_x = (\pi/\lambda_{SAW})(2A_{SAW}/m)^{1/2} = 1.5$ THz (see Supplementary Note 4 in ref. 18). The aspect ratio of the 2D confinement is thus $\omega_x/\omega_y = 0.21$.

The potential being entirely determined, the effective electron temperature $T$ is the only free parameter to be chosen for good agreement with the experimental data, as shown by the solid lines in Fig. 3b–e using $T = 25$ K. This value is also consistent with the one extracted from the barrier-height dependence of the thermally activated hopping rate between the two wells of the quartic potential, as estimated in Supplementary Note 2.

## Data availability

All data from this study are included in the paper and its Supplementary Information. Source data are provided with this paper.

## Code availability

All codes used in this study are available from the corresponding authors on request.

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

**Acknowledgements** We acknowledge fruitful discussions with U. A. Wiedemann, D. Zavickis and B. Sacépé during the preparation of this manuscript. This project has received financing from the European Union Horizon 2020 research and innovation programme under grant agreement no. 862683, 'UltraFastNano'. C.B., H.S. and J.S. acknowledge financing from Agence Nationale de la Recherche under the France 2030 programme, reference ANR-22-PETQ-0012. C.B. acknowledges financial support from the Agence Nationale de la Recherche project QUABS ANR-21-CE47-0013-01. J.W. acknowledges the European Union Horizon 2020 research and innovation programme under Marie Skłodowska-Curie grant agreement no. 754303. M.A. acknowledges the Marie Skłodowska-Curie Actions cofund QuanG grant no. 101081458, financed by the European Union and the programme QuanTEdu-France no. ANR-22-CMAS-0001 France 2030. L.M. acknowledges the programme QuanTEdu-France no. ANR-22-CMAS-0001 France 2030. T.V. acknowledges financing from the French Laboratory of Excellence project 'LANEF' (ANR-10-LABX-0051). A.D.W. and A.L. thank the German Research Foundation (Deutsche Forschungsgemeinschaft, DFG) through ML4Q EXC 2004/1 - 390534769, the BMBF-QR.X project 16KISQ009 and the DFH/UFA project CDFA-05-06. E.P., R.S. and V.K. have been supported by the Latvian Quantum Initiative within European Union Recovery and Resilience Facility project no. 2.3.1.1.i.0/1/22/I/CFLA/001 and grant no. lzp2021/1-0232 from the Latvian Council of Science. Views and opinions expressed are those of the authors only and do not necessarily reflect those of the European Union or the granting authority. Neither the European Union nor the granting authority can be held responsible for them.

**Author contributions** J.S. performed the experiment, with expert help from J.W. and technical support from S.O., T.V., M.A., L.M., C.G. and M.U. J.W. fabricated the sample. E.P. performed the cumulant calculations and R.S. the Monte Carlo simulations, with guidance from V.K. A.L. and A.D.W. provided the high-quality GaAs/AlGaAs heterostructure. J.S., E.P., C.B., V.K. and H.S. wrote the manuscript, with feedback from all authors. H.S. and C.B. supervised the experimental work. V.K. conceptualized the theoretical interpretation. C.B. initiated the project.

**Competing interests** The authors declare no competing interests.

**Additional information**
**Correspondence and requests for materials** should be addressed to Vyacheslavs Kashcheyevs or Hermann Sellier.

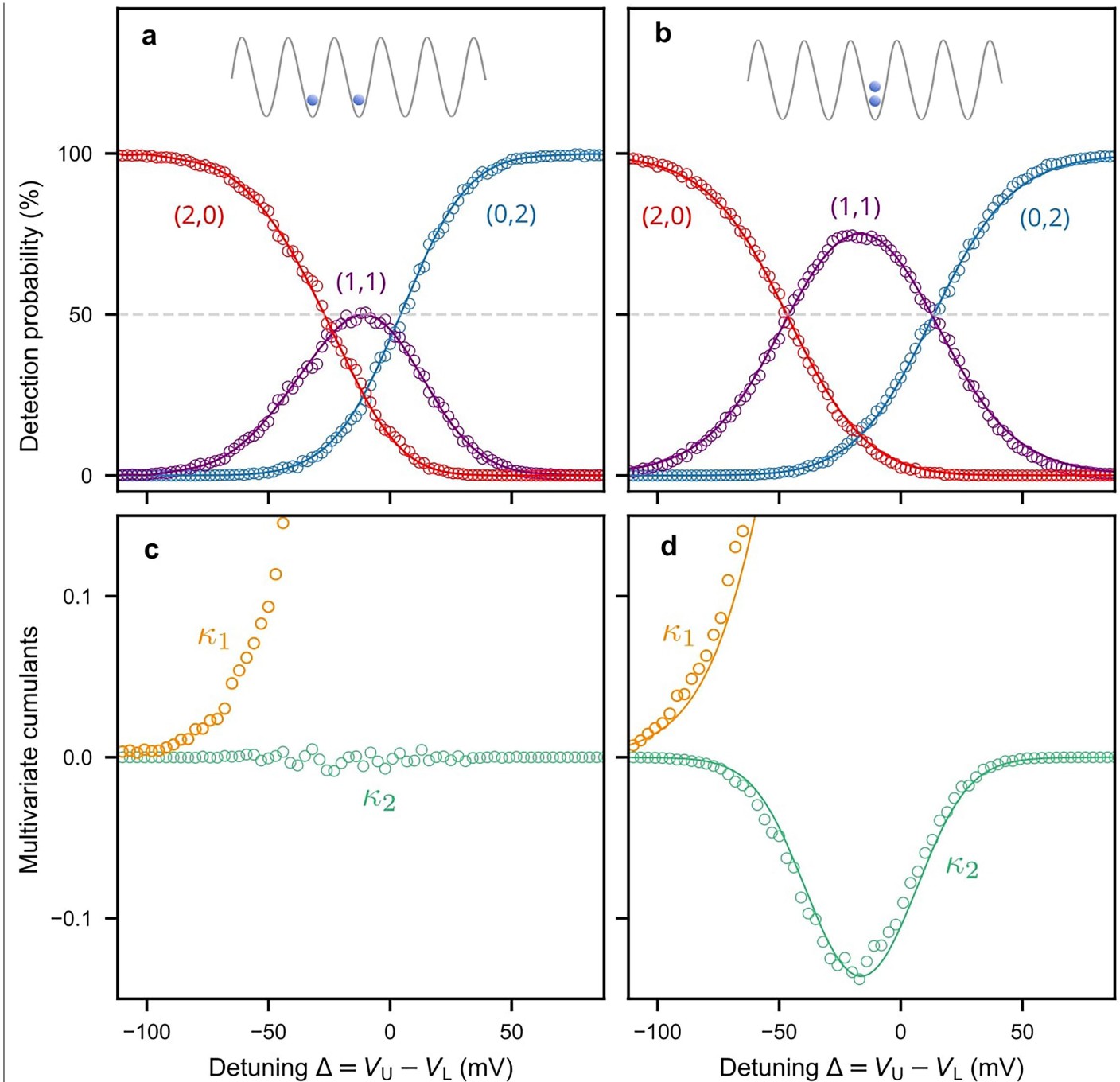

**Extended Data Fig. 1 | Experimental data for partitioning of *N* = 2 electrons.**
**a**, Partitioning probabilities when the two electrons are distributed in two different minima and are uncorrelated. **b**, Partitioning probabilities when both electrons are in the same SAW minimum and are interacting. **c**,**d**, The multivariate cumulants corresponding to **a** and **b**, respectively. Lines in **a** are reconstructions using single-electron partitioning data. Lines in **b** and **d** are fitting curves from the Ising model using the parameters given in Extended Data Table 1.

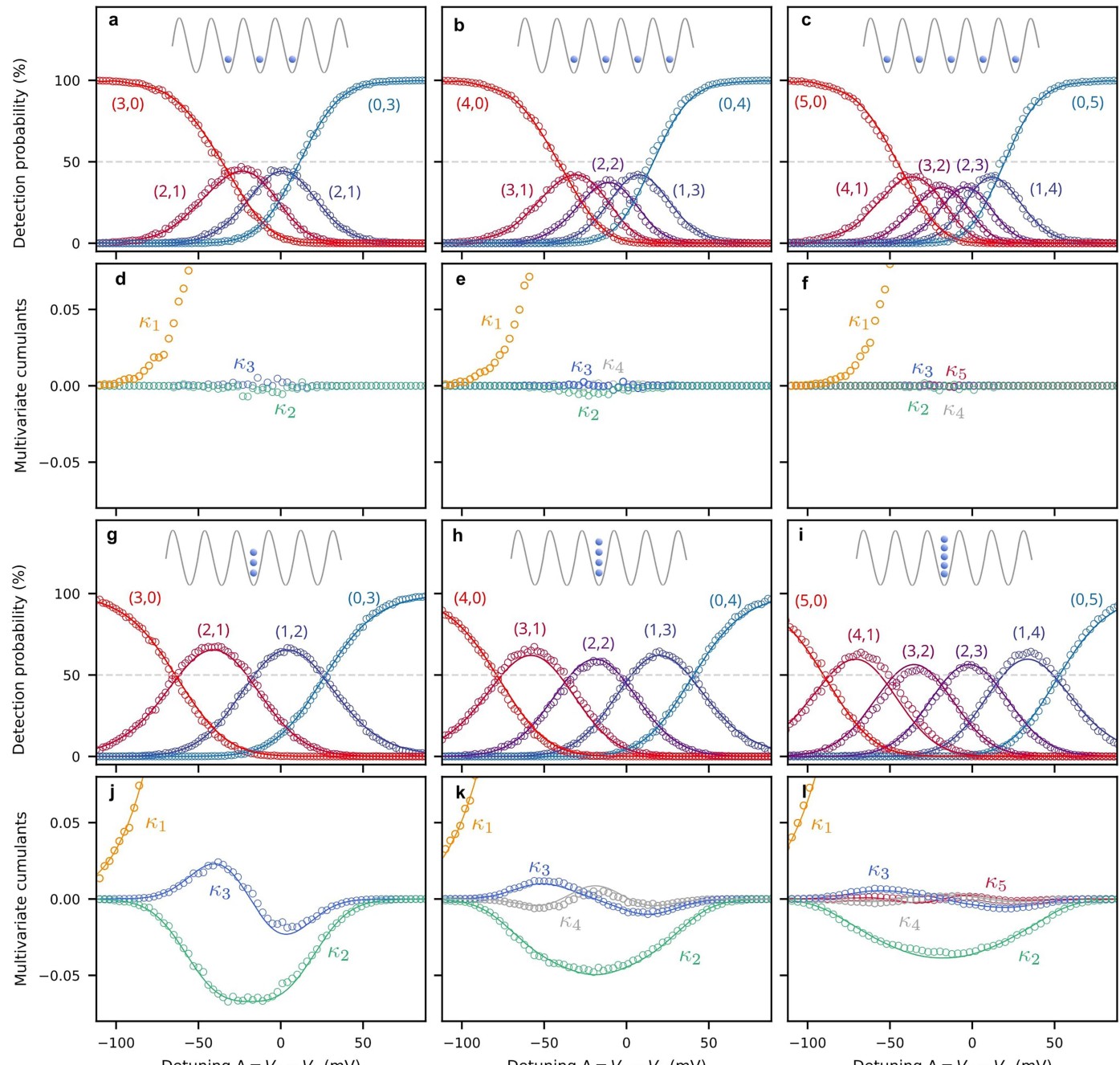

**Extended Data Fig. 2 | Experimental data for partitioning of N = 3, 4 and 5 electrons. a–f,** The partitioning probabilities and the corresponding multivariate cumulants for electrons distributed across different SAW minima (uncorrelated electrons). **g–l,** The same quantities when all electrons are placed in the same SAW minimum (interacting electrons). Lines in **a–c** are reconstructions using single-electron partitioning data. Lines in **g–l** are fitting curves from the Ising model using the parameters given in Extended Data Table 1.

**Extended Data Table 1 | Fitting parameters of the Ising model for partitioning statistics**

| $N$ | $\Delta_0$ (mV) | $\alpha/k_BT$ (mV$^{-1}$) | $U/k_BT$ | $T/T_N$ |
|---|---|---|---|---|
| 1 | −12.5 | 0.064 | – | – |
| 2 | −16.4 | 0.064 | 1.22 | 0.82 |
| 3 | −18.5 | 0.064 | 0.94 | 0.71 |
| 4 | −19.0 | 0.064 | 0.80 | 0.63 |
| 5 | −17.8 | 0.064 | 0.70 | 0.57 |

The parameters $\Delta_0$, $\alpha/k_BT$ and $U/k_BT$ correspond to the partitioning experiments. The last column shows corresponding values of $T/T_N=2k_BT/UN$.