## [Peer Review File · Nature]

Evidence of Coulomb liquid phase in few-electron droplets

Corresponding Author: Dr Hermann Sellier

Version 1:

Reviewer comments:

Referee #1

(Remarks to the Author)

The authors demonstrate the partitioning of the electron droplet in the mesoscopic system. They form the electron droplet up to 5 electrons, and transport it using the surface acoustic wave (SAW) technique. The droplet is partitioned at the quantum point contact. The authors measure the partitioning probabilities and analyze them using counting statistics or cumulants. Since the SAW potential minima behave as moving quantum dots, electrons are well localized in the potential minimum of SAW. Depending on the distribution of electrons in the SAW minima, electrons either interact with each other or do not.

When all electrons are localized in the same potential, they interact strongly with each other and form a strongly correlated state (or Coulomb liquid). The authors show the signature of strong correlation in the Coulomb liquid using the partitioning probabilities and more clearly in multivariate cumulants. The authors map this system onto the Ising model on a complete graph (an all-to-all interacting Ising model) and explain the cumulants.

Recently many theoretical and experimental papers have discussed particle colliders in condensed matter systems, whose goal is to identify quasiparticles with exotic statistics. However this paper demonstrates a fundamentally different way to explore the strongly correlated state by partitioning the strongly correlated systems, which mimics the breaking-up matter in high-energy physics. Although there are several limitations, such as limited number of electrons, this work opens a novel avenue for studying the phases of interacting electrons. As such, I believe this novelty warrants publication in Nature.

Below I list the questions and comments to improve the paper:

1. Some of the comparisons with QCD seem unnecessary. For example, the comparison between the Coulomb gas-liquid transition in this work and the transition from quark-gluon plasma to hadronic fluid seems unnecessary. Rather than the comparison I would suggest focusing on analyzing the Coulomb gas-liquid transition in more detail.

2. The author measures the electron number in the SAW potential minima using a QPC. To measure the charge using the QPC, sufficient time is needed to collect the tunneled particles, and this time must be shorter than the time scale defined by the speed of the SAW. If this can be explained with a simple comparison of a few numbers, could you include it in the main text?

3. It is obvious that placing more electrons in the SAW is very difficult. However, is it possible to overcome this issue by adding more sources as the authors demonstrated in the paper? Specifically, if there are additional sources such as S3 (in addition to S1 and S2), could more than 5 electrons be placed?

4. In Figure 2, the authors compare multivariate cumulants from experimental data with theoretical results derived from the Ising model on a complete graph. However, there are no theoretical results provided for partitioning probabilities. Is it difficult to obtain theoretical predictions for the partitioning probabilities?

Referee #2

(Remarks to the Author)

This manuscript presents an investigation of the states of what authors call a droplet, which is a small region in a semiconductor material that contains a few electrons. These droplets are split into two, and the statistics of the resultant number of electrons are studied. They introduce multivariate cumulants in their analysis, and find a universal behaviour of the liquid state as they vary the number of electrons in the droplets.

They introduce the subject by a fascinating analogy with the phase transitions between hadrons/quark-gluon plasma. This illustrates the importance in understanding the nature of droplets of elementary particles. However, the authors' experiments are conducted at much lower energy states. No dramatic change (compared to hadron collisions) in the electron state is observed.

The phase transition between the liquid state and gas state is gradual, rather than abrupt. It is characterised by a crossover temperature T_c , which depends on the Coulomb strength and number of electrons in the droplet, according to the Ising model described. The value of T_c is not explicitly quoted in the manuscript. It may be that it is difficult to deduce the values from the experimental data available. They use a droplet temperature $T = 25$ K with the value of T/T_N provided in their partitioning simulation. This implies that the crossover temperature is around 30 - 40 K.

The most striking result is that the behaviour of the multivariate cumulants is universal, i.e. the results from different numbers scale onto the same functional curve. This implies that the behaviour of the few electron droplets, while their physical state may be complicated, can be expressed by a relatively simple statistical model. It also shows the concreteness of their experiments with careful device tuning.

While their sophisticated experimental technique and theoretical analysis are to be praised, it should be noted that they did not explore the phase transition experimentally, i.e. whether they can turn their droplet state into the gas phase. This could have put their methodology under the test.

In terms of the readability of the manuscript, it is well written with detailed information. While the details of the cumulants and Ising model are left to the Method section and the Supplementary Material, the math and their interpretation in the main manuscript may still be challenging for the general readers of Nature who may not be familiar with the concept.

In terms of the statistics of measurement data, no error bars were provided in the data plots, but I do not think they are necessary for the nature of the data provided. The trend is clear in the data plots and there is no reason to believe that the data analysis is skewed by the lack of uncertainty validation.

In my opinion, the authors work should appeal to many in the field of quantum electron transport. They conducted experiments in a spatially moving framework (droplets carried by surface acoustic waves), but I can imagine that similar experiments could be carried out in a gate-defined static system with fast gate operations. Then the same statistical analysis treatment can be applied to investigate the state of (static) droplets (or people may prefer to call "quantum dots").

The manuscript also contains an argument that appeals to general readers, in the analogy with quark-gluon plasma in hadron collision experiments. However, the implication of their experiments, in comparison with the dramatic event of hadron collision, that would appeal to general readers is not clear to me. The authors finishes their first paragraph with "Our electron-droplet collider provides critical insight into the interplay of confinement and interaction effects in small electron systems and highlights a new way to study engineered states of matter." I suggest the authors to clarify what they mean by the "critical insight" they obtained, which should appeal to general readers. Otherwise, this sentence could be read as that this manuscript is about a new experimental technique that would appeal only to researchers working on small electron systems. If I understand correctly, the "critical insight" they found is that the few electron droplets that they created are in a strongly-correlated liquid state, rather than in a "hot" gas state. If authors could clarify the importance of this finding to people in other disciplines, that would improve the final sentence in the first paragraph, and I would be satisfied to recommend the manuscript for publication in Nature.

There is one more suggestion that I would like to make, relating to the same sentence, though semantically. The authors call their device "electron-droplet collider". Their device is designed so that electrons can come from two sources and meet at the central barrier gate, so this looks like a "collider", in which two electron droplets collide. However, they argue that these electrons forget where they have come from after they merge into one droplet. The manuscript focuses on what happens after one droplet is split into two at the exit Y junction. Therefore, we could argue that the "collision" part is irrelevant. I would like the authors to clarify why they use the term "collider". If "collision" event is not important as in my argument above, then they should change the word as it gives a false impression to readers.

Referee #3

(Remarks to the Author)

The manuscript 'Evidence of Coulomb liquid phase in few-electron droplets' by J. Shaju and co-workers reports on a joined

experimental and theoretical study of strong Coulomb interactions in few-electron droplets, which are probed with an on-chip electron-collider.

The authors use quantum dots (QDs) with nearby charge detectors to prepare electron droplets, which are injected into a channel region through a surface acoustic wave (SAW). After interacting in the channel, the droplet is guided to a voltage-controlled two-way splitter and the total charge of the fragments is probed in either end of the splitter. The authors find evidence for anti-bunching in the counting statistics, indicative of strong Coulomb repulsion. The experimental findings are in good agreement with an almost parameter-free effective Ising model.

I find the experimental approach to studying strong electronic correlations using a solid-state analog of a high-energy particle collider very interesting and highly relevant for studying strongly interacting fermionic systems. Further, I am convinced that the idea will appeal to a broad audience.

I find that the experimental part of the work was carried out with great rigor and reported on with a high amount of detail. In the supplement, the authors demonstrate control over the loading and detection of the electron droplets. Further they demonstrate a consistent behaviour of the counting statistics for different total numbers of electrons and several distributions of electrons into different SAW pockets. The effective Ising model seems to nicely agree with the experimental data.

Overall, I support the publication of the work in Nature given the high relevance, also to a broader audience, the careful design of the study and the excellent experimental and theoretical work. The manuscript is clearly written, and I have only a few minor points that the authors should address:

1.) To me, it is not fully clear what the advantage of choosing a moving electron droplet is. Could one simply form a large QD with 5 electrons and non-adiabatically change the confinement potential to form a double quantum dot with gate-controlled detuning and get the same result?

2.) Is there a deeper or more subtle physical difference between having $N=4$ with two electrons in two different SAW minima or having $N=2$ with the two electrons being in the same pocket? Can I not simply regard the first case as two independent shots of the second case?

Minor things:

1.) The effective temperature assumed for modelling the droplet was chosen to be $T=25\text{K}$, this is slightly different from the experimental determination of the droplets' temperature. Is there a reason why the authors do not simply use the experimentally determined value for the modelling?

2.) The experimental values in Fig. 4b (small rectangles) are very hard to see, and their color is hard to determine. I suggest that the authors try to improve on the representation.

Manuscript: “Evidence of Coulomb liquid phase in few-electron droplets” by Shaju *et al.*

We appreciate the valuable feedback of the reviewers. We take note of the overall sentiment shared by Reviewers #1 and #2, that analogy with quark-gluon deconfinement transition, although captivating, risks misleading the reader as the corresponding interactions are very different and a phase transition is not directly explored in our experiments. We have modified our manuscript accordingly to tone down this analogy but keep the focus on the concepts and methods that we transpose to a new context. The most significant change is the reduction of the particle physics exposition in the Introduction as follows (changes are indicated in green):

Breaking-up matter into pieces and studying the statistics of fragments is one of the basic epistemic strategies in physics. Arguably the most exquisite pursuit of this strategy is the success of high-energy particle colliders in discovering and quantifying the fundamental types of matter within the Standard Model of elementary particles. In the studies of strong interaction, relativistic ion collisions are used to induce the deconfinement of the nuclear matter (composed of correlated hadrons) into a hot plasma of more fundamental particles (quarks and gluons) [13]. The statistical fluctuations of collision products carry rich information about the collective dynamics [9]. In particular, measurements of high-order cumulants have been used [2, 14] to pinpoint the critical point in the phase diagram of quantum chromodynamics (QCD) [15, 16].

We address the other comments of the reviewers in a detailed point-by-point response below.

Response to Reviewer #1

Referee #1 (Remarks to the Author):

The authors demonstrate the partitioning of the electron droplet in the mesoscopic system. They form the electron droplet up to 5 electrons, and transport it using the surface acoustic wave (SAW) technique. The droplet is partitioned at the quantum point contact. The authors measure the partitioning probabilities and analyze them using counting statistics or cumulants.

Since the SAW potential minima behave as moving quantum dots, electrons are well localized in the potential minimum of SAW. Depending on the distribution of electrons in the SAW minima, electrons either interact with each other or do not. When all electrons are localized in the same potential, they interact strongly with each other and form a strongly correlated state (or Coulomb liquid). The authors show the signature of strong correlation in the Coulomb liquid using the partitioning probabilities and more clearly in multivariate cumulants. The authors map this system onto the Ising model on a complete graph (an all-to-all interacting Ising model) and explain the cumulants.

Recently many theoretical and experimental papers have discussed particle colliders in condensed matter systems, whose goal is to identify quasiparticles with exotic statistics. However this paper demonstrates a fundamentally different way to explore the strongly correlated state by partitioning the strongly correlated systems, which mimics the breaking-up matter in high-energy physics. Although there are several limitations, such as limited number of electrons, this work opens a novel avenue for studying the phases of interacting electrons. As such, I believe this novelty warrants publication in Nature.

Below I list the questions and comments to improve the paper:

1. Some of the comparisons with QCD seem unnecessary. For example, the comparison between the Coulomb gas-liquid transition in this work and the transition from quark-gluon plasma to hadronic fluid seems unnecessary. Rather than the comparison I would suggest focusing on analyzing the Coulomb gas-liquid transition in more detail.

1.1) In addition to removing unnecessary details of QCD physics from the introduction, we have reformulated the paragraph opening the section “Universal signatures of a Coulomb liquid” to put more emphasis on the Coulomb plasma properties, and in particular on the Coulomb liquid state distinguished by its low compressibility. Only the common physical features and modelling assumptions between ion collisions and our droplet partitioning are retained (quench of a thermal state after fragmentation, high-T and low-T connected by a smooth crossover):

The relevant state of matter for our electron droplets is a one-component Coulomb plasma [41] which can undergo a temperature-driven crossover from a Coulomb gas at $T > T_c$ to Coulomb liquid at $T < T_c$. The crossover temperature T_c is determined by the competition between entropy and Coulomb repulsion energy, and is characterized by the dimensionless plasma parameter $\Gamma^{(pl)} \sim T_c/T$. In more conventional transport experiments, where statically-confined electrons are connected to external reservoirs, transition to a Coulomb liquid can manifest itself as an energy gap on

the order of $k_B T_c / N$ for particle addition (Coulomb blockade in quantum dots) or as temperature saturation of compressibility of a two-dimensional electron gas [42].

Here we rely solely on the finite- N counting statistics to estimate the state of our Coulomb plasma, building an analogy with relativistic ion collisions used to study the phase diagram of QCD. In particular, at low baryonic densities [43], the transition from quark-gluon plasma at temperatures $T > T_c^{QCD}$ to hadronic fluid at $T < T_c^{QCD}$ is not a sharp phase transition but rather a smooth crossover [44, 45]. Freeze-out of fluctuations (due to quench of equilibrium during expansion [16]) determines the cumulants in the number of produced hadrons, which have been used to estimate $k_B T_c^{QCD} \sim 170$ MeV [2]. Unlike in QCD, where particles are created from the vacuum, the equilibrium ensemble for our Coulomb droplets is canonical, as the number of electrons N is conserved in collisions with the Y-junction.

2. The author measures the electron number in the SAW potential minima using a QPC. To measure the charge using the QPC, sufficient time is needed to collect the tunneled particles, and this time must be shorter than the time scale defined by the speed of the SAW. If this can be explained with a simple comparison of a few numbers, could you include it in the main text?

1.2) We think that there has been a misunderstanding about the detection technique used in our experiment. After partitioning at the Y-junction, the electrons transported by the SAW are captured in the detector quantum dots while the SAW continues its propagation. The captured electrons are stored for a sufficiently long time to be measured with high accuracy using the nearby QPC sensors. This charge measurement is done after the SAW train has traversed the entire sample structure. To make this point clearer, we have improved the wording of the following sentence:

By recording the QPC current I_{QPC} of each QD before and after the experiment, the precise number of released electrons (source QD) and captured electrons (detector QD) is measured.

3. It is obvious that placing more electrons in the SAW is very difficult. However, is it possible to overcome this issue by adding more sources as the authors demonstrated in the paper? Specifically, if there are additional sources such as S3 (in addition to S1 and S2), could more than 5 electrons be placed?

1.3) One can indeed add more sources to increase the number of electrons, but one could also improve the design of the quantum dots to emit (source) or trap (detector) more electrons in presence of the SAW. Another limitation is the number of electrons that can be transported in the moving quantum dot, which depends on the SAW amplitude. We estimate that we could transfer up to 10 electrons with proper design and presently available SAW amplitude.

4. In Figure 2, the authors compare multivariate cumulants from experimental data with theoretical results derived from the Ising model on a complete graph. However, there are no theoretical results provided for partitioning probabilities. Is it difficult to obtain theoretical predictions for the partitioning probabilities?

1.4) The theoretical modelling of the droplet partitioning is independent of the representation, being in terms of probabilities or in terms of cumulants, since these two quantities are directly related. For completeness, we have added the theoretical fits on the experimental probabilities in Extended Data Figure 1 for $N=2$ and Extended Data Figure 2 for $N=3,4,5$.

Response to Reviewer #2

Referee #2 (Remarks to the Author):

This manuscript presents an investigation of the states of what authors call a droplet, which is a small region in a semiconductor material that contains a few electrons. These droplets are split into two, and the statistics of the resultant number of electrons are studied. They introduce multivariate cumulants in their analysis, and find a universal behaviour of the liquid state as they vary the number of electrons in the droplets.

They introduce the subject by a fascinating analogy with the phase transitions between hadrons/quark-gluon plasma. This illustrates the importance in understanding the nature of droplets of elementary particles. However, the authors' experiments are conducted at much lower energy states. No dramatic change (compared to hadron collisions) in the electron state is observed.

2.1) We believe the essential conceptual import from hadron-liquid/quark-gluon-plasma high-energy physics to low-temperature electronics that we have put forward is two-fold: studying collective effects in collisions of composite droplets (heavy nuclei as droplets of hadron matter in contrast to proton-to-proton collisions) and aiming to put the results on a universal phase diagram. It is indeed true that our experiments did not explore experimentally a phase change of electronic matter (we do hope follow-up studies will). In response to these accurate observations by Reviewer #2 and a similar criticism by Reviewer #1 above, we have reduced the exposition of QCD physics in the Introduction (see changes at the top of our response letter) and refocused the discussion of the strongly-correlated liquid limit on the relevant Coulomb physics, while trying not to lose the fresh perspective brought by the high-energy collider analogy (see response 1.1).

The phase transition between the liquid state and gas state is gradual, rather than abrupt. It is characterised by a crossover temperature T_c , which depends on the Coulomb strength and number of electrons in the droplet, according to the Ising model described. The value of T_c is not explicitly quoted in the manuscript. It may be that it is difficult to deduce the values from the experimental data available. They use a droplet temperature $T = 25$ K with the value of T/T_N provided in their partitioning simulation. This implies that the crossover temperature is around 30 - 40 K.

2.2) The crossover temperature T_c indeed can be identified with the Néel temperature T_N . The exact definition for the crossover temperature is nevertheless model specific, hence the distinct notation. We choose not to emphasize the specific value of T_N in the text because it depends on the non-universal confinement strength in our particular realization (unlike T_c^{QCD} which is a universal scale for the fundamental strong interaction) and can be easily obtained from the numbers quoted. The calculation $T_N \sim 40$ K by the Reviewer is therefore completely correct. We also note that the value $T = 25$ K estimated from the simulation of the multi-electron partitioning data is backed up by two other estimates described in detail in section 2.4 of the Supplementary Material. To make sure this point is not missed by the readers, we have added the following remark at the end of section "Effective Ising model":

... the only adjustable parameter is the effective temperature of the droplet, $T = 25$ K (consistent with other estimates, see Supplementary Note 2).

The most striking result is that the behaviour of the multivariate cumulants is universal, i.e. the results from different numbers scale onto the same functional curve. This implies that the behaviour of the few electron droplets, while their physical state may be complicated, can be expressed by a relatively simple statistical model. It also shows the concreteness of their experiments with careful device tuning.

While their sophisticated experimental technique and theoretical analysis are to be praised, it should be noted that they did not explore the phase transition experimentally, i.e. whether they can turn their droplet state into the gas phase. This could have put their methodology under the test.

2.3) We are glad that the powerful nature of the evidence for scaling is appreciated. We also agree that controlled driving of the liquid-gas crossover is an important next step in experimental validation of the methodology. To explore the phase diagram, one would have to vary the ratio T/T_c which means varying either the electronic temperature T or the crossover temperature T_c . For example, reaching a lower electronic temperature would require a semiconductor platform with less potential fluctuations than in doped GaAs/AlGaAs heterostructure (source of excitations during the flight), while reducing the Néel temperature would require a reduced charging energy of the moving quantum dot. By changing the voltages on the upper and lower side gates, one could possibly change the capacitance to ground of the moving dot, and thereby change its charging energy e^2/C_{total} , but changing the side gate voltages by a large amount while keeping a high transfer fidelity is unfortunately not possible. Alternatively, varying the barrier height in the central channel could be another path to explore.

In terms of the readability of the manuscript, it is well written with detailed information. While the details of the cumulants and Ising model are left to the Method section and the Supplementary Material, the math and their interpretation in the main manuscript may still be challenging for the general readers of Nature who may not be familiar with the concept.

2.4) We have strived to keep the technical arguments in the main text as self-sufficient as feasible. This indeed may be a strain for the readers less-versed in the corresponding concepts. To remedy the situation, we have added short sentences explaining the intuitive meaning in a few places of the main text where the argument gets

most abstract. To help the reader in understanding the meaning of the cumulants and model parameters, we have added the following sentences:

In section "Multivariate cumulants":

$\kappa_2 = \langle\langle T_1 T_2 \rangle\rangle < 0$ means the choice, that one electron makes, tends to be opposite to what the other electrons do.

In section "Universal signatures of a Coulomb liquid":

In statistical physics, the liquid phase is characterized by low compressibility as it resists changes in particle number. Thus the strongly-correlated limit, derived here from the condition of vanishing fluctuations in the thermodynamic limit, $\langle\langle n^2 \rangle\rangle/N \rightarrow 0$, corresponds to an incompressible fluid.

In section "Effective Ising model":

U is the energy cost for two spins to be parallel (for two electrons to exit on the same side of the Y-junction) with a positive U making configurations with opposite spins preferable (antiferromagnetic coupling).

In terms of the statistics of measurement data, no error bars were provided in the data plots, but I do not think they are necessary for the nature of the data provided. The trend is clear in the data plots and there is no reason to believe that the data analysis is skewed by the lack of uncertainty validation.

2.5) We appreciate that the reviewer backs our judgement on the presentation of statistical uncertainty. The uncertainty intervals for the experimentally-estimated counting probabilities have been taken into account in Ising model parameter estimation. We have added a brief description of this uncertainty estimation to the Methods, in a new section called "Statistical uncertainty", and the following sentence in the caption of Fig. 2:

Error bars on probabilities are smaller than symbol size (see Methods).

In my opinion, the authors work should appeal to many in the field of quantum electron transport. They conducted experiments in a spatially moving framework (droplets carried by surface acoustic waves), but I can imagine that similar experiments could be carried out in a gate-defined static system with fast gate operations. Then the same statistical analysis treatment can be applied to investigate the state of (static) droplets (or people may prefer to call "quantum dots").

2.6) The partitioning of a multi-electron droplet into two fragments could indeed be realised using a static dot equipped with a central barrier forming a double dot. However, the advantage of using a moving quantum dot is that the SAW transports the two fragments towards well-separated detectors capable of unambiguously measuring the number charge of each fragment. If one would use a static dot, the two fragments would stay nearby each other after the partitioning of the droplet. After rising the central barrier to separate the dot into two isolated regions, the two fragments would be separated by only a few 100 nm and each fragment would be capacitively coupled to the two charge sensors (although with different capacitance) which would significantly complicate the discrimination between the multiple configurations of charge states in case of partitioning a droplet with several electrons. On the other hand, we expect our experiment to inspire new studies on partitioning a multi-electron droplet into two fragments using a static dot with a central barrier forming a double dot. This approach will pave the way for exploring correlated electron systems of finite size in their quantum ground state.

The manuscript also contains an argument that appeals to general readers, in the analogy with quark-gluon plasma in hadron collision experiments. However, the implication of their experiments, in comparison with the dramatic event of hadron collision, that would appeal to general readers is not clear to me. The authors finish their first paragraph with "Our electron-droplet collider provides critical insight into the interplay of confinement and interaction effects in small electron systems and highlights a new way to study engineered states of matter." I suggest the authors to clarify what they mean by the "critical insight" they obtained, which should appeal to general readers. Otherwise, this sentence could be read as that this manuscript is about a new experimental technique that would appeal only to researchers working on small electron systems. If I understand correctly, the "critical insight" they found is that the few electron droplets that they created are in a strongly-correlated liquid state, rather than in a "hot" gas state. If authors could clarify the importance of this finding to people in other disciplines, that would improve the final sentence in the first paragraph, and I would be satisfied to recommend the manuscript for publication in Nature.

2.7) We agree that the original version of the concluding sentence of the introductory paragraph is too focused on small electron systems and does not help general readers appreciate the broader implications of our findings. The "critical insight" for electron droplets was indeed meant to be discovering characteristics of a liquid, but the main surprise (as correctly identified in the comments of the Reviewer #2 above the response 2.3) is in getting a robust signature with so few particles. To make the broader implications clear, we have changed the ending of the introductory paragraph as follows:

Our electron-droplet scattering experiments illustrate how coordinated behavior emerges via interactions of a mere handful of elementary constituents. Studying similar signatures in other physical platforms such as cold-atom simulators [4, 11] or collections of anyonic excitations [8, 12] may help identify emergence of exotic phases and, more broadly, advance understanding of matter engineering.

There is one more suggestion that I would like to make, relating to the same sentence, though semantically. The authors call their device "electron-droplet collider". Their device is designed so that electrons can come from two sources and meet at the central barrier gate, so this looks like a "collider", in which two electron droplets collide. However, they argue that these electrons forget where they have come from after they merge into one droplet. The manuscript focuses on what happens after one droplet is split into two at the exit Y junction. Therefore, we could argue that the "collision" part is irrelevant. I would like the authors to clarify why they use the term "collider". If "collision" event is not important as in my argument above, then they should change the word as it gives a false impression to readers.

2.8) Indeed our use of the word "collision"/"collider" is confusing, since it has been used in two meanings: collision of two droplets as a technique to prepare the initial state with maximal N vs the subsequent collision of the synthesised droplet with the Y-junction. Moreover, the technical term "collider" in particle physics tends to be used exclusively for two counterpropagating beams and not for fixed-target experiments. Hence we follow the Reviewer's recommendation and remove the word "collider" which was used twice in the manuscript:

- "Our electron-droplet **collider** provides..." has been modified to: "Our electron-droplet *scattering experiments* ..."
- "On-chip multi-electron **collider**" has been modified to: "On-chip multi-electron *splitter*".

Response to Reviewer #3

Referee #3 (Remarks to the Author):

The manuscript 'Evidence of Coulomb liquid phase in few-electron droplets' by J. Shaju and co-workers reports on a joined experimental and theoretical study of strong Coulomb interactions in few-electron droplets, which are probed with an on-chip electron-collider.

The authors use quantum dots (QDs) with nearby charge detectors to prepare electron droplets, which are injected into a channel region through a surface acoustic wave (SAW). After interacting in the channel, the droplet is guided to a voltage-controlled two-way splitter and the total charge of the fragments is probed in either end of the splitter. The authors find evidence for anti-bunching in the counting statistics, indicative of strong Coulomb repulsion. The experimental findings are in good agreement with an almost parameter-free effective Ising model.

I find the experimental approach to studying strong electronic correlations using a solid-state analog of a high-energy particle collider very interesting and highly relevant for studying strongly interacting fermionic systems. Further, I am convinced that the idea will appeal to a broad audience.

I find that the experimental part of the work was carried out with great rigor and reported on with a high amount of detail. In the supplement, the authors demonstrate control over the loading and detection of the electron droplets. Further they demonstrate a consistent behaviour of the counting statistics for different total numbers of electrons and several distributions of electrons into different SAW pockets. The effective Ising model seems to nicely agree with the experimental data.

Overall, I support the publication of the work in Nature given the high relevance, also to a broader audience, the careful design of the study and the excellent experimental and theoretical work. The manuscript is clearly written, and I have only a few minor points that the authors should address:

1.) To me, it is not fully clear what the advantage of choosing a moving electron droplet is. Could one simply form a large QD with 5 electrons and non-adiabatically change the confinement potential to form a double quantum dot with gate-controlled detuning and get the same result?

3.1) The partitioning of a multi-electron droplet into two fragments could indeed be realized using a static dot equipped with a central barrier forming a double dot. The results would be the same if the electrons would have the same effective temperature. As Reviewer #2 makes a similar observation, we explain in answer 2.6 the advantages of our setup.

2.) Is there a deeper or more subtle physical difference between having $N=4$ with two electrons in two different SAW minima or having $N=2$ with the two electrons being in the same pocket? Can I not simply regard the first case as two independent shots of the second case?

3.2) Indeed, there is no difference between $2e$ ($N=2$) and $2e/2e$ ($N=4$) if the electrons in different minima do not interact. The configuration $2e/2e$ ($N=4$) shown in Fig. 2b is analysed in detail in section 6 of the Supplementary Material (see Fig. S8). It confirms that electrons in different minima are independent, and also illustrates that different orders of multivariate cumulants can distinguish the number of particles contributing to correlations.

Minor things:

1.) The effective temperature assumed for modelling the droplet was chosen to be $T=25\text{K}$, this is slightly different from the experimental determination of the droplets' temperature. Is there a reason why the authors do not simply use the experimentally determined value for the modelling?

3.3) We could have used the experimentally-determined temperature of about 30 K (from barrier height and detuning dependencies of single-electron partitioning) but we find it interesting to obtain the temperature from matching the multi-electron partitioning data with the Monte-Carlo simulations, which gives 25 K, and then to check if this value is consistent with the 30 K obtained using a very different characterization method. Each estimate makes its own simplifying assumptions and the degree of consistency between the results is a measure of the overall robustness of our microscopic modelling. This discussion can be found in section 2.4 of the Supplementary Material.

2.) The experimental values in Fig. 4b (small rectangles) are very hard to see, and their color is hard to determine. I suggest that the authors try to improve on the representation.

3.4) The experimental values shown by small boxes were indeed too small. We improved them in the new version of Fig. 4b and added a label on each box indicating the particle number N .

Additional modifications

Apart from the changes detailed above, three new references have been added:

[11] J. C. Halimeh, M. Aidelsburger, F. Grusdt, P. Hauke, and B. Yang, *Cold-atom quantum simulators of gauge theories*, *Nature Physics* 21, 25 (2025)

[12] J. Nakamura, S. Liang, G. C. Gardner, and M. J. Manfra, *Direct observation of anyonic braiding statistics*, *Nature Physics* 16, 931 (2020)

[42] J. P. Eisenstein, L. N. Pfeiffer, and K. W. West, *Compressibility of the two-dimensional electron gas: Measurements of the zero-field exchange energy and fractional quantum hall gap*, *Phys. Rev. B* 50, 1760 (1994).